# Value Function Decomposition in Markov Recommendation Process

## ABSTRACT

Recent advances in recommender systems have shown that user-system interaction essentially formulates long-term optimization problems, and online reinforcement learning can be adopted to improve recommendation performance. The general solution framework incorporates a value function that estimates the user's expected cumulative rewards in the future and guides the training of the recommendation policy. To avoid local maxima, the policy may explore potential high-quality actions during inference to increase the chance of finding better future rewards. To accommodate the stepwise recommendation process, one widely adopted approach to learning the value function is learning from the difference between the values of two consecutive states of a user. However, we argue that this paradigm involves an incorrect approximation in the stochastic process. Specifically, between the current state and the next state in each training sample, there exist two separate random factors from the stochastic policy and the uncertain user environment. Original TD learning under these mixed random factors may result in a suboptimal estimation of the long-term rewards. As a solution, we show that these two factors can be separately approximated by decomposing the original temporal difference loss. The disentangled learning framework can achieve a more accurate estimation with faster learning and improved robustness against action exploration. As empirical verification of our proposed method, we conduct offline experiments with online simulated environments built based on public datasets.

## KEYWORDS

Recommender Systems, Reinforcement Learning, Markov Decision Process

**ACM Reference Format:**
Anonymous Author(s). 2018. Value Function Decomposition in Markov Recommendation Process. In *Proceedings of (WWW '25)*. ACM, New York, NY, USA, 14 pages. https://doi.org/XXXXXXX.XXXXXXX

## 1 INTRODUCTION

Recommender systems play a crucial role in enhancing user experience across a variety of online platforms such as e-commerce, news, social media, and micro-video platforms. Their primary objective is to filter and recommend content that aligns with users' interests

A note.

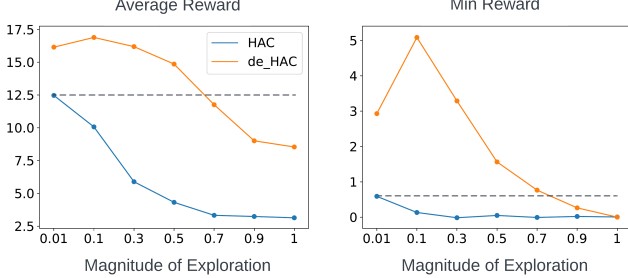

**Figure 1: The proposed decomposition method on the RL backbone (i.e. HAC) in KuaiRand dataset improves the overall performance and is more robust to exploration of recommendation actions.**

and preferences, improving user engagement with the platform. Early studies considered this as a ranking problem and built collaborative filtering solutions [20, 21, 34] aimed at minimizing the errors between item-wise labels and the ranking score prediction. Later approaches found that sequential modeling [17, 18, 36] of user histories can better capture the dynamics of user interest and offer more accurate predictions of the future. In recent studies, many recommendation scenarios have shown that the learning target should also go beyond immediate feedback and extend to the future influence, in which reinforcement learning (RL) methods [2, 49] can further improve the long-term cumulative reward and achieve state-of-the-art recommendation performance.

The fundamental idea behind RL-based recommendation methods is considering the user-system interaction sequence as a Markov Decision Process (MDP) [31, 32, 35] so that each recommendation action only depends on the current user state and optimize the long-term performance. Specifically, each context-aware user request consists of the user's static profile features and dynamic interaction history, which is later encoded as the user state. Between the consecutive user states, the recommendation policy first takes the current state as input and outputs a recommendation list (or item) as the action, then the user environment receives this action and generates user feedback that will determine the immediate reward and the transition toward the next state. This interaction between the policy and the user forms a full cycle in the Markov Recommendation Process (MRP). Then the goal is usually formulated as the maximization of the cumulative reward which represents the long-term performance of the policy. In other words, RL-based methods optimize the policy with the total effect in the future as a target label, which is different from traditional learning-to-rank methods [27] that only optimize the policy with immediate feedback. Then, the key to effective guidance of the policy is finding an accurate value function that approximates the expected long-term reward for sampled actions in given states. To accommodate the stepwise samples

in recommendation problems and rapidly adapt the dynamic user interests in the online learning environment, a temporal difference (TD) learning technique is adopted [38, 49] that either minimizes the error between the two consecutive state evaluation (denoted as Value-based TD) or minimize that between the two consecutive state-action pairs (denoted as action's Quality-based TD), as illustrated in Figure 2-a.

Though they are effective, we find that it is challenging to obtain a stable and accurate value function in online RL due to the severe exploration-exploitation trade-off [4, 10, 26] in recommender systems. On one hand, TD learning may achieve better value function accuracy when the policy's exploration of actions is restricted to a small variance (which may work well in simple scenarios with a small item candidate pool), but it also reduces the chance of finding better actions and has a higher chance being trapped in local maxima. On the other hand, the policy may increase the magnitude of action exploration to find potentially better policies, but this also makes it harder for stable and accurate value function learning due to the increased variance. In this paper, we argue that one of the key reasons that limit the accuracy of the value function is the mixed view of the two random factors in the MRP: the policy's **random action exploration** and the **stochastic user environment**. As we will illustrate in section 3.2 and Appendix A, mixing the two random factors would introduce a negative effect on stepwise TD learning. As a consequence, the resulting value function becomes suboptimal and limits the effectiveness of exploration.

To address the aforementioned limitations, we propose to decompose the standard TD learning paradigm of the value function into two separate sub-problems with respect to each random factor, as shown in Figure 2-b. In the first sub-problem, our primary focus is developing an accurate approximation of the user state's long-term utility, mitigating the influences from the random policy. In contrast, the second sub-problem focuses on refining a precise function for the state-action pair, which captures the recommendation actions' effectiveness, excluding the influence of the inherent randomness of the user environment. We show that the decomposed objectives bound the original TD learning objective, and the exclusion of unrelated random factors potentially speeds up the learning process. As empirical verification, we show the superiority of our solution in finding better policies through online evaluation of simulated environments. Meanwhile, the resulting framework becomes more robust to action exploration as exemplified in Figure 1. In extreme cases where the policy "overexplores" the action space, the proposed method can still effectively optimize the value function while the baseline crashes in terms of recommendation performance.

In summary, our contributions are as follows:

- We specify the challenge of suboptimal TD learning under the mixed random factors between policy action exploration and stochastic user environment.
- We propose a decomposed TD learning framework that separately addresses the two random factors and empirically shows its superiority in online RL-based solution.
- We verify that the proposed decomposition technique provides more robust performance under action exploration

and a faster learning process across multiple TD-learning-based methods.

## 2 RELATED WORK AND PROBLEM DEFINITION

### 2.1 Reinforcement Learning for Recommendation

The RL-based recommendation system [1, 38, 49] operates within the Markov Decision Process (MDP) framework, aiming to optimize cumulative rewards which reflects the long-term user satisfaction. While tabular-based methods [29] can optimize an evaluation table in simple settings, they are constrained to a small fixed set of state-action pairs. For larger action space and state space, studies have found solutions with value-based methods [33, 39, 48, 51], policy gradient methods [6, 7, 13, 14, 23, 42], and actor-critic methods [5, 43–46, 49, 50]. Among existing methods, the temporal difference (TD) technique [38] has been widely used to learn and optimize long-term rewards due to its stepwise learning framework that well-suits the recommendation task and online learning environment. Our method also aligns with this paradigm. Despite the efficacy of TD learning, reinforcement learning encounters new challenges in accommodating recommender systems, including exploration in combinatorial state/actions space [10, 16, 25, 26], dealing with unstable user behavior [3, 8], addressing heterogeneous user feedback [5, 9], and managing multi-task learning [11, 28, 40].

Additionally, in the realm of general reinforcement learning [38], there are several works that described possible alternatives for TD learning [30, 37] in specific scenarios. One of the works that is closer in methodology is the Dueling DQN [41]. It proposes a way to decompose the $Q$ function into a value function and advantage function so that one can learn a $V$ function in the Q-learning framework.

### 2.2 Problem Formulation

In this section, we present the Markov Recommendation Process (MRP) for online RL. Assume a candidate pool of N items denoted by $I$ and assume a pre-defined reward function $r(\cdot)$ for the observed user feedback. Then, the MRP components are:

- $S$: the continuous representation space of the user state, and each state $s_t$ encodes the user and context information upon the recommendation request at step $t$.
- $A$: the action space corresponds to the possible recommendation lists. For simplicity, we consider the list of fixed size $K$ so the action space is $A = I^K$.
- $r(s_t, a_t)$: the immediate reward that captures the user feedback for the recommendation action $a_t \in A$ on user state $s_t \in S$. In this paper, we denote $r_t = r(s_t, a_t)$ .
- $\pi : S \rightarrow A$, the recommendation policy that outputs an item or a list of items as an action for each request, and we assume that the policy applies random action exploration in the online learning setup.
- $P : S \times A \rightarrow S$, the stochastic state transition function where the randomness only comes from the user environment. In other words, the recommendation problem has a stochastic partially

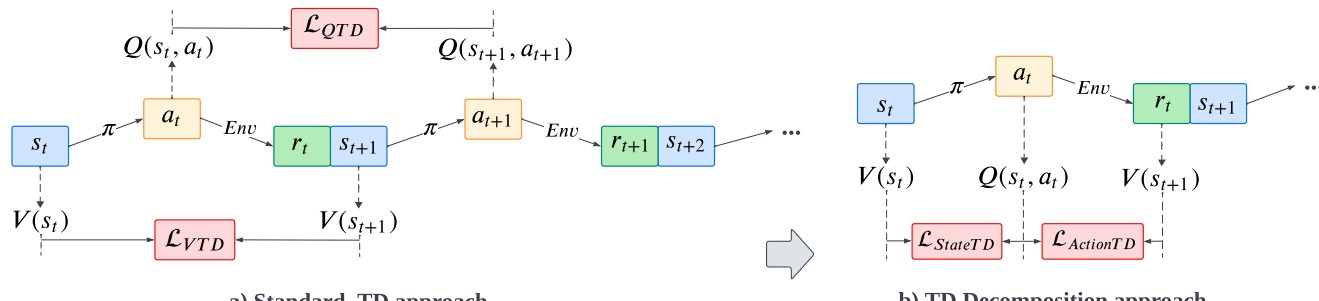

a) Standard TD approach

b) TD Decomposition approach

**Figure 2: The general Markov Recommendation Process. Standard TD approaches (left) either adopt $Q$-based or $V$-based TD. Our solution (right) decomposes the learning into two objectives for random policy and stochastic user environment respectively.**

observable user environment, and the next-state distribution of $P(s_{t+1}|s_t, a_t)$ is assumed unknown.

Then, for each stepwise interaction cycle, a training sample collects the information as a tuple $\mathcal{D}_t = (s_t, a_t, r_t, s_{t+1}, d)$ where $d \in \{0, 1\}$ represents whether the session ends after taking action $a_t$. Following the intuition of long-term performance optimization, the **Goal** is to learn a policy that can generate an action $a_t$ at any step $t$ that maximizes the user's expected cumulative reward over the interactions in the future:

$$\mathbb{E}[r_t] = \mathbb{E}[\sum_{i=0}^{\infty} \gamma^i r_{t+i}] \tag{1}$$

where $\gamma \in [0, 1]$ is the discount factor that balances the focus of immediate reward and the long-term rewards, and the expectation term implicitly includes the two random sampling factors i.e. policy and user environment. Note that in the online RL setting, we ignore the user's leave-and-return behavior (and the influence of signal $d$) by the end of each session, and assume an infinite horizon of the MRP as reflected in Eq.(1). Additionally, the user state encoder usually adopts neural networks to encode the user profile and context features and uses sequential models to dynamically encode the user interaction history in practice. In this work, we consider the detailed encoder design as complementary work and focus on the reasoning of the learning framework.

## 3 METHOD

### 3.1 Reinforcement Learning with Temporal Difference

Directly optimizing Eq.(1) requires the sampling of the user's trajectories, but this is impractical for recommendation scenarios with large numbers of users and items. As an alternative, Temporal Difference (TD) learning can naturally accommodate the stepwise online learning environment of the recommender system, taking advantage of dynamic programming(DP) and Monte Carlo methods(MC). Specifically, for each given data sample $\mathcal{D}_t$ it defines a value function $V(s_t)$ that estimates Eq.(1) at any step $t$. Then the temporal difference between two consecutive states can be captured by the value function estimator:

$$V(s_t) = \mathbb{E}_{a_t|s_t} \mathbb{E}_{s_{t+1}, r_t|s_t, a_t} \left[ r_t + \gamma V(s_{t+1}) \right] \tag{2}$$

where the first expectation considers the random policy and the second expectation corresponds to the stochastic user environment. Then we can (approximate it with sampling and) optimize the difference between $V(s_t)$ and $V(s_{t+1})$ through the error to the observed immediate reward, which derives the standard value-based TD loss:

$$\mathcal{L}_{\text{VTD}} = \left( r_t + \gamma V(s_{t+1}) - V(s_t) \right)^2 \tag{3}$$

Similarly, the difference between state-action pairs also has the corresponding approximation:

$$Q(s_t, a_t) = \mathbb{E}_{s_{t+1}, r_t|s_t, a_t} \left[ r_t + \gamma \mathbb{E}_{a_t|s_t} \left[ Q(s_{t+1}, a_{t+1}) \right] \right] \tag{4}$$

which derives the following Q-based TD loss:

$$\mathcal{L}_{\text{QTD}} = \left( r_t + \gamma Q(s_{t+1}, a_{t+1}) - Q(s_t, a_t) \right)^2 \tag{5}$$

These two types of functions describe different segments of the MDP as illustrated in Figure 2-a. While the learned value function $V$ estimates the expected performance of the observed user state, the learned $Q$ function estimates the expected performance of an action on a given state.

Ideally, when the value function or the Q function is well-trained and accurately approximates the expected cumulative reward, we can use them to guide the policy either through the advantage boosting loss as in A2C [19]:

$$\mathcal{L}_{\text{policy}} = -A_t \log \pi(a_t|s_t)$$
$$A_t = r_t + \gamma V(s_{t+1}) - V(s_t) \tag{6}$$

where the larger advantage $A_t$ an action generates, the more likely this action gets selected; or we can optimize the policy in an end-to-end manner through expected reward maximization as in DDPG:

$$\mathcal{L}_{\text{policy}} = -Q(s_t, a_t)$$
$$a_t \sim \pi(\cdot|s_t) \tag{7}$$

where a larger $Q$ estimation of the action induces a higher chance of selection.

### 3.2 The Challenge of Mixing Random Factors

Though the aforementioned TD learning has been proven effective, the overall framework essentially ignores the effect of the mixed random factors from policy and the user environment as described

in section 1. Specifically, the randomness in the user environment merely depends on the user's decision which is conditionally independent from the policy, but it directly affects the observed reward for a given state-action pair. For example, the user may still skip the recommended item when something else draws the attention, even if the item is attractive to the user. In contrast, the policy's action is a controllable random factor in terms of the exploration magnitude. It is conditioned on the given state, but only indirectly affects the observed reward with the existence of stochastic users.

However, TD learning in Eq.(3) and Eq.(5) does not distinguish these two factors which results in suboptimal estimation. Particularly, we can define the random difference brought by the user as $\Delta_u$ which partially explains the error between the estimation of next-state's $V$ and the $Q$ of the current state-action pair:

$$r_t + \gamma V(s_{t+1}) = Q(s_t, a_t) + \Delta_u \tag{8}$$

which instantiates the statistical relation:

$$Q(s_t, a_t) = \mathbb{E}_{s_{t+1}, r_t | s_t, a_t} \left[ r_t + \gamma V(s_{t+1}) \right] \tag{9}$$

Similarly, we can define $\Delta_\pi$ as the difference brought by the policy's random action which partially explains the error between the estimation the state value $V$ and $Q$ of the state-action pair:

$$Q(s_t, a_t) = V(s_t) + \Delta_\pi \tag{10}$$

which instantiates the statistical relation:

$$V(s_t) = \mathbb{E}_{a_t | s_t} \left[ Q(s_t, a_t) \right] \tag{11}$$

Then, combining Eq.(3) and Eq.(8), the value-based TD learning for the value function $V$ becomes:

$$\mathcal{L}_{\text{VTD}} = \left( Q(s_t, a_t) + \Delta_u - V(s_t) \right)^2 \tag{12}$$

where the existence of $\Delta_u$ (which is conditionally independent from $Q$) makes it harder to reach the correct estimation of Eq.(11). Furthermore, during policy optimization such as Eq.(6), the advantage term will also include this user random factor (i.e. $A_t = Q(s_t, a_t) + \Delta_u - V(s_t)$). This may misguide the policy because of the user's influence in $\Delta_u$. Similarly, combining Eq.(5) with Eq.(10), the Q-based TD learning for the $Q$ function becomes:

$$\mathcal{L}_{\text{QTD}} = \left( r_t + \gamma V(s_{t+1}) + \gamma \Delta_\pi - Q(s_t, a_t) \right)^2 \tag{13}$$

where the existence of $\Delta_\pi$ (which is independent of the previous stochastic user state transition) introduces extra noise for the approximation of Eq.(9). This may potentially downgrade the effectiveness of the $Q$ function (e.g. using Eq.(7)) and become reluctant to guide the policy learning.

In both cases, the inaccurate TD learning is suboptimal and requires more sampling efforts to approach a valid approximation, which potentially results in slower and harder training. Furthermore, when adopting action exploration in online RL, one may have to restrict the exploration magnitude to a relatively low level in order to keep $\delta_\pi$ small and increase the accuracy of the estimation. However, this scarifies the model's exploration ability and has a lower chance of reaching global maxima.

## 3.3 Exclude Irrelevant Random Factors in TD Decomposition

As a straightforward derivation from the analysis in section 3.2, we propose to eliminate the irrelevant terms during training. The resulting framework consists of two separate learning objectives for random policy and stochastic user environment respectively.

The first objective optimizes the $Q$ function with the $V$ function fixed (with stopped gradient):

$$\mathcal{L}_{\text{actionTD}} = \left( r(s_t, a_t) + \gamma V(s_{t+1}) - Q(s_t, a_t) \right)^2 \tag{14}$$

which directly matches Eq.(9). This objective focuses on learning a correct estimate of $Q(s_t, a_t)$, which is conditioned on the sampled action. In other words, $\mathcal{L}_{\text{actionTD}}$ optimizes $Q$ to capture $\Delta_\pi$ and eliminate the effect of $\Delta_u$ by error minimization. The second objective optimizes the $V$ function with the $Q$ function fixed:

$$\mathcal{L}_{\text{stateTD}} = \left( V(s_t) - Q(s_t, a_t) \right)^2 \tag{15}$$

which directly matches the goal of Eq.(11). This objective focuses on learning the correct value function $V(s_t)$ of a given state without the influence of a random action exploration. In other words, $\mathcal{L}_{\text{stateTD}}$ optimizes $V$ to capture $\Delta_u$ and eliminate the effect of $\Delta_\pi$ through error minimization.

Combining the two objectives, we form the **TD Decomposition** framework that simultaneously optimizes $Q$ and $V$ as shown in Figure 2-b, and both functions can be approximated by neural networks. While the state learning objective $\mathcal{L}_{\text{stateTD}}$ uses $Q$ as the label for $V$, the $\mathcal{L}_{\text{actionTD}}$ uses immediate reward $r_t$ and $V$ as targets for $Q$. The combined learning framework is theoretically more accurate due to the removed noise from irrelevant random factors and consistently bounds the original TD learning. In comparison, optimizing the standard $\mathcal{L}_{\text{VTD}}$ and $\mathcal{L}_{\text{QTD}}$ sometimes misguide the learning of $V$ and $Q$. We present the details of these analyses in Appendix A.

In addition to the improved accuracy, the decomposed TD also has several extra advantages:

- Because the decomposition removes the irrelevant terms in each separate learning task, the corresponding $V$ and $Q$ can learn from more accurate signals with fewer samples. In other words, we expect a **faster learning** under this new framework as we will verify in section 4.2.2.
- When increasing the exploration of action, the $\Delta_\pi$ is only captured by $Q(s_t, a_t)$ in Eq.(14). The large variance of $\Delta_\pi$ does not affect the learning of $V$ since it is removed from Eq.(15). Intuitively, this would help improve the **robustness** against action exploration as we provide empirical evidence in section 4.3.2.
- The framework will learn both $V$ and $Q$ functions which can adapt to TD-based methods that uses either Eq.(3) or Eq.(5). This means that this decomposition is a **general technique** that can benefit a wide range of RL-based recommender systems, including but not limited to A2C and DDPG.

## 3.4 Action Discrepancy and Debiased Decomposition

In online RL, another challenge that may affect the accuracy of TD learning is the discrepancy between the action distribution in the

past and the present, especially when the policy frequently changes along with user dynamics and continuous training. Without loss of generality, let $\pi(a_t|s_t)$ represent the likelihood of generating $a_t$ using the current policy and let $p(a_t|s_t)$ represent the observed likelihood from the past policy when the sample is collected. Consider the correct expected loss as:

$$\mathbb{E}_{a_t \sim \pi}[\mathcal{L}_{\text{stateTD}}] \tag{16}$$

taking the derivative and the minimization point with zero gradient corresponds to:

$$2 \int_{a_t} \pi(a_t|s_t)(V(s_t) - Q(s_t, a_t)) = 0$$
$$\Rightarrow V(s_t) \int_{a_t} \pi(a_t|s_t) = \int_{a_t} \pi(a_t|s_t) Q(s_t, a_t)) \tag{17}$$
$$\Rightarrow V(s_t) = \mathbb{E}[Q(s_t, a_t)] = V^*$$

where $V^*$ represents the correct value estimation. Yet, the sampled action in the past does not necessarily follow the distribution of $\pi$, which explains the aforementioned discrepancy. As a countermeasure in our decomposed TD learning, we include an extra debias term $\beta$ for the state TD:

$$\mathcal{L}_{\beta-\text{stateTD}} = \beta \Big( V(s_t) - Q(s_t, a_t) \Big)^2$$
$$\beta = \frac{\pi(a_t|s_t)}{p(a_t|s_t)} \tag{18}$$

which is theoretically derived from the following transformation:

$$\mathbb{E}_{a_t \sim \pi}[\mathcal{L}_{\text{stateTD}}] = \int_{a_t} \pi(a_t|s_t) \big( V(s_t) - Q(s_t, a_t) \big)^2$$
$$= \int_{a_t} p(a_t|s_t) \frac{\pi(a_t|s_t)}{p(a_t|s_t)} \big( V(s_t) - Q(s_t, a_t) \big)^2 \tag{19}$$
$$= \mathbb{E}_{a_t \sim p}[\mathcal{L}_{\beta-\text{stateTD}}]$$

Intuitively, this debias term would help refine the learning of $V$ towards a closer estimation of the correct target $V^*$ of the current policy even when the sample comes from a policy in the past.

## 4 EXPERIMENTS

In this section, we illustrate the experimental support for our claims through the evaluation of simulated online learning environments. We summarize our research focus as follows:

- Verify the correctness and faster convergence of our decomposition method by recommendation performance comparison with stepwise TD counterpart.
- Verify that the proposed TD decomposition is more robust to action exploration.
- Analyze the behavior of the state TD loss and action TD loss and the stability of the combined optimization.

## 4.1 Experimental Settings

*4.1.1 Datasets and Online Simulator.* We include three public datasets in our experiments: MovieLens-1M[15], Amazon(book)[22] and KuaiRand1K[12]. The ML1M dataset contains one million user ratings of movies, while KuaiRand1K is a dataset that includes multi-behavior user interaction records with short videos sampled

for one thousand users. Note that the traditional offline evaluation in the recommender system is not well-suited for online RL methods since they do not provide the estimation of dynamically changing environment and labels for unseen interaction sequences. Instead, we preprocess the datasets and construct the simulated environment for online learning similar to that in KuaiSim[47]. Both datasets were cleaned by removing users/items with fewer than 10 interactions and reconstructed records chronologically. In order to generate realistic user feedback, a user response model is trained to estimate the probability of a user's click based on their dynamic interaction history and static profile features. During online RL, the user simulator will produce immediate feedback (of user clicks) according to this model and serve as the interactive environment. The reward design follows the KuaiSim system which considers a reward of 1.0 for a click and -0.2 for a missing click. The maximum episode depth is limited to 20 by the temper-based user leave model which maintains a user's budget of temper, and the budget decreases during the online interactions until it reaches a threshold and triggers the leaving of the user.

*4.1.2 Evaluation Protocol.* After preparing datasets and their corresponding online simulators, we can use the simulated user environment to engage in training of online RL models. Empirically, all tested methods converge within 30,000 steps and we evaluate their average performance in the last 100 episode steps. As main evaluation metrics, we include the average total **reward** (without discount) of a user session and session **depth** as accuracy indicators. For extreme cases, we include the **minimum reward** metric of user sessions in each batch sample. We would identify the superior performance as the aforementioned accuracy metrics have higher values. In addition, to observe the stability of the method, we also include the **reward variance** for each batch of samples.

*4.1.3 Baselines.* We implemented the following baselines to provide a comparison in our evaluation:

- Supervision (Non-RL): a supervised learning method similar to [18], which uses transformer to encode user history and neural networks to encode user profile. The item-wise score is a dot product between user encoding and item encoding, and we optimize it through binary cross-entropy loss.
- A2C [19]: a family of actor-critic RL methods that combine the policy gradient optimization with Eq.(6) and V-learning approaches with Eq.(3).
- DQN [31]: a model-free RL method that uses a deep neural network to approximate the Q-value function with Eq.(5). DQN employs an epsilon-greedy strategy to balance exploration and exploitation during the learning process.
- DDPG [24]: an actor-critic framework that optimizes the critic with Eq.(5), and optimizes the policy actions in the continuous action space with Eq.(7) with Gaussian-based exploration.
- HAC [26]: an advanced version of DDPG specifically designed for recommendation. It extends the actor-critic framework for request-level scenarios and uses a vectorized hyper action to represent each item list. HAC includes additional action space regularization and item-wise supervision to further improve performance and learning stability.

 

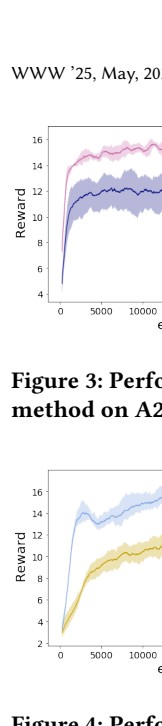

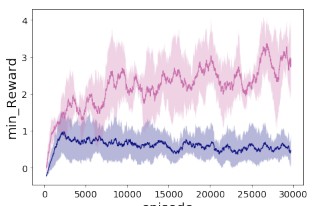

**Figure 3: Performance between original and decomposed TD method on A2C in KuaiRand dataset.**

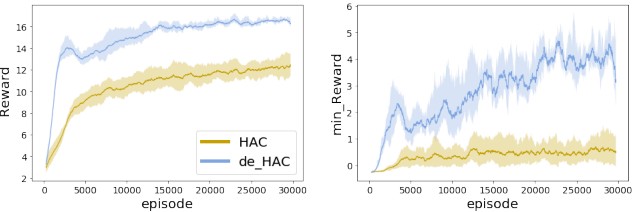

**Figure 4: Performance between original and decomposed TD method on HAC in KuaiRand dataset.**

- SQN [43]: SQN augments existing recommendation models with an additional reinforcement learning output layer that serves as a regularizer, allowing the model to focus on specific rewards.
- Dueling DQN [41] (D-DQN): a model-free RL algorithm that extends the Q-Learning algorithm to deal with the problem of learning in continuous action spaces. The key innovation of Dueling DQN is the introduction of a dueling network architecture that separates the computation of state-value function as $Q(s_t, a_t) = V(s_t) + A(s_t, a_t)$, which achieves the value estimation under the Q-learning objective.

For all methods, we implement an experience replay buffer for the online learning process and the exploration of action will directly influence the sample distribution in the buffer. For frameworks that use TD learning (i.e. A2C, DQN, DDPG, and HAC), we apply the proposed TD decomposition to verify its effectiveness across various RL backbones. D-DQN cannot integrate TD decomposition since it is already a decomposition method. To ensure fair comparison, we adopt the same neural network structure across $V$ functions, and the same structure across $Q$ as well. The user states are obtained using the same structure as the user encoder in the Supervision baseline, and all RL methods use this same state encoder design. For reproduction of our empirical study, we provide implementation and training details in our released source code [1].

## 4.2 Main results

*4.2.1 Recommendation Performance:* For each experiment of all models, we conducted five rounds of online training with different random seeds and reported the average results in Table (1). We can see that the A2C, DDPG, and HAC can consistently improve performance over supervision baselines, indicating the superiority of RL methods that can optimize long-term user rewards. The DDPG

---

[1] https://anonymous.4open.science/r/TD_Decomposition

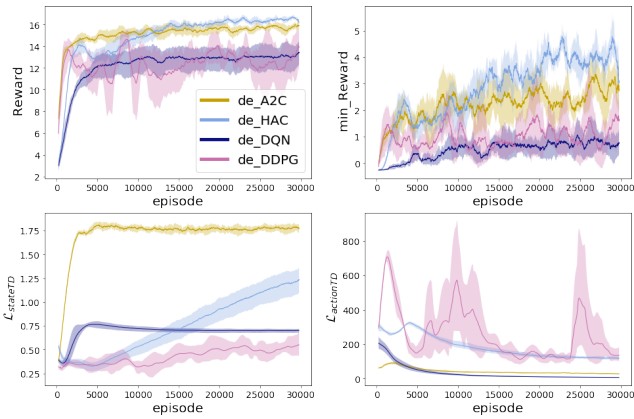

**Figure 5: $\mathcal{L}_{stateTD}$ and $\mathcal{L}_{actionTD}$ curves for TD decomposition methods**

only improves the results in KuaiRand but is inferior to supervision in ML1M, which might indicate that the ML1M environment is easier as a recommendation task. The Dueling DQN method learns V and Advantage simultaneously and generates Q for the TD learning process. However, this decomposition does not solve the problem of mixing random factors and may introduce extra learning costs to achieve the same level of accuracy. As a result, its performance appears to be suboptimal compared to other advanced RL methods.

In general, the proposed TD decomposition demonstrates stronger performance than the original TD, and this observation is consistent across all four backbones (A2C, DQN, DDPG, and HAC) and across both datasets. Specifically, in the ML1M environment, the decomposed methods exhibit slight improvements in rewards as 2.5% for A2C, 1.2% for DQN, 26.1% for DDPG, and 5.2% for HAC; and improvements in depths as 1.9% for A2C, 1.0% for DQN, 20.9% for DDPG and 4.3% for HAC. And the improvements in DDPG and HAC are statistically significant. In KuaiRand1K, the relative improvement of decomposition is 33.6% for A2C, 25.1% for DQN, 8.2% for DDPG, and 35.4% for HAC in terms of total rewards; and 26.0% for A2C, 18.7% for DQN, 5.5% for DDPG, and 27.7% for HAC. All improvements are statistically significant (student-t test with $p < 0.05$). Note that the overall improvement of TD decomposition is larger in KuaiRand than in ML1M, which indicates a harder recommendation environment. This difference might be related to the fact that short-video platforms involve more dynamics of users' intensive interactions, compared with movie recommendations.

*4.2.2 Faster Learning of TD Decomposition.* To further illustrate the training behavior of the TD decomposition method, we plot the learning curves of the two most effective baselines (i.e. A2C and HAC) and compare them with TD decomposition counterparts in Figure 3 and Figure 4. We can see that the TD decomposition achieves a faster and better reward boost in the beginning and the converged point reveals consistently better performance. In the extreme cases illustrated by the minimum reward plot, the value functions from the original TD learning become reluctant to guide the policy in the later training process, but the decomposition methods achieve continuous improvement over time indicating a

**Table 1: Online simulation performance of all methods and their correspongding decomposition.The better performances compared with native and decomposed in bold and the best in Underline.**

| Model | Total Reward in ML1M | | Total Reward in KuaiRand | | Total Reward in Amazon | |
|---|---|---|---|---|---|---|
| | Original | Decomposed | Original | Decomposed | Original | Decomposed |
| Non-RL | 15.97 ±(2.21) | - | 10.28 ±(3.78) | - | - | - |
| Dueling DQN | 15.83 ±(0.31) | - | 10.79 ±(4.38) | - | 10.43 ±(3.27) | - |
| A2C | 17.19 ±(0.34) | **17.62** ±(0.23) | 11.91 ±(0.90) | **15.91** ±(0.36) | 11.24 ±(0.78) | **13.11** ±(0.92) |
| DQN | 15.95 ±(0.53) | **16.14** ±(0.42) | 10.74 ±(4.68) | **13.44** ±(1.41) | 10.36 ±(3.60) | **11.94** ±(1.41) |
| DDPG | 13.52 ±(1.83) | **17.05** ±(1.01) | 12.86 ±(1.65) | **13.78** ±(1.59) | 10.99 ±(1.85) | **11.58** ±(1.52) |
| HAC | 16.89 ±(1.80) | **17.76** ±(0.42) | 12.47 ±(1.00) | **16.89** ±(0.52) | 12.17 ±(0.63) | **13.31** ±(1.02) |
| SQN | 16.33 ±(0.45) | **16.88** ±(0.38) | 11.22 ±(0.76) | **15.42** ±(0.70) | 6.94 ±(0.53) | **11.74** ±(0.83) |

**Table 2: The effect of action exploration in HAC. $\sigma$ represents the magnitude of action exploration. The better performances compared with native and decomposed in bold and the best in Underline.**

| $\sigma$ | Total Reward in ML1M | | Total Reward in KuaiRand | | Total Reward in Amazon | |
|---|---|---|---|---|---|---|
| | Original | Decomposed | Original | Decomposed | Original | Decomposed |
| 1 | 11.95 ±(5.28) | **17.70** ±(0.45) | 3.14 ±(0.12) | **8.54** ±(0.22) | 1.20 ±(0.18) | **2.01** ±(0.28) |
| 0.9 | 12.48 ±(5.31) | **17.66** ±(0.34) | 3.24 ±(0.23) | **9.01** ±(0.30) | 1.17 ±(0.12) | **2.14** ±(0.17) |
| 0.7 | 13.35 ±(5.18) | **17.38** ±(0.68) | 3.33 ±(0.28) | **11.76** ±(0.14) | 1.25 ±(0.22) | **2.82** ±(0.37) |
| 0.5 | 14.32 ±(4.43) | **17.58** ±(0.27) | 4.32 ±(0.87) | **14.86** ±(0.49) | 1.31 ±(0.33) | **4.17** ±(0.49) |
| 0.3 | 15.48 ±(2.52) | **17.59** ±(0.59) | 5.89 ±(0.75) | **16.19** ±(0.25) | 2.30 ±(1.11) | **9.31** ±(0.68) |
| 1e-1 | 16.89 ±(1.80) | **17.76** ±(0.42) | 10.07 ±(1.09) | **16.89** ±(0.52) | 6.75 ±(0.84) | **12.97** ±(0.86) |
| 1e-2 | 17.06 ±(1.03) | **17.37** ±(0.94) | 12.47 ±(1.00) | **16.15** ±(0.59) | 12.17 ±(0.63) | **13.31** ±(1.02) |

more accurate guidance with continuous exploration. We present more details about these learning curves with longer training steps in Appendix C.

Note that the decomposition framework is a general technique that can accommodate any RL methods that engage TD learning, but the policy learning and action exploration might still behave differently even with an improved value function. Figure 5 shows the comparison of different RL backbones with the TD decomposition. Except for the DDPG backbone is relatively unstable, all other RL methods achieve stable learning for both $\mathcal{L}_{\text{actionTD}}$ and $\mathcal{L}_{\text{stateTD}}$.

## 4.3 Ablation

*4.3.1 Impact of Learning Rates on Two-Step TD.* Recall that in our TD decomposition method, we perform two separate learning objectives for $V$ and $Q$. This means that we can separately manipulate the learning rates for $V$ and $Q$. In Figure 6 we show the effect of learning rate of $V$ with learning rate of $Q$ fixed, and in Figure 7 we show the effect of learning rate of $Q$ with the learning rate of $V$ fixed. We can generally observe the under-fitting and over-fitting on the two sides of the reward performance. And there are several patterns that worth noticing:

- the reward performance is negatively correlated with the reward variance;
- $\mathcal{L}_{\text{actionTD}}$ and $\mathcal{L}_{\text{stateTD}}$ behave in opposite directions when changing the learning rate of $V$, which mean that aligning $V$ with $Q$ under more restrictions may loss the ability of

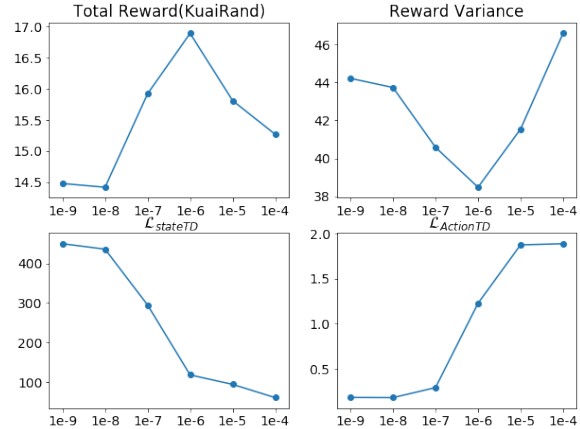

**Figure 6: Effect of TD decomposition with HAC on KuaiRand dataset. X-axis correspond to the learning rate for $V$ learning.**

expectation approximation and introduce larger error in the learning of $Q$;
- $\mathcal{L}_{\text{actionTD}}$ and $\mathcal{L}_{\text{stateTD}}$ behave in the same directions when changing the learning rate of $Q$, which means that achieving a more accurate $Q$ estimation results in more accurate $V$ estimation.

We provide extended results with more details in appendix D

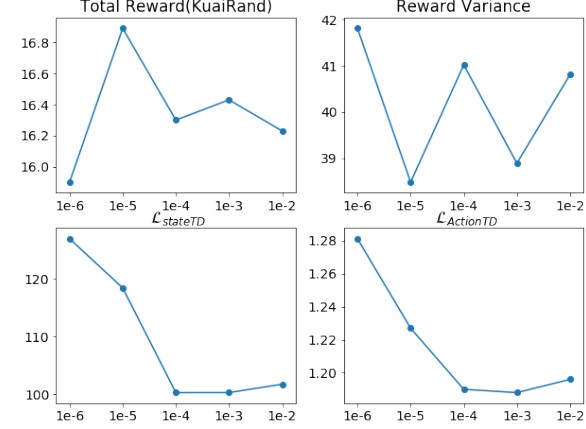

**Figure 7: Effect of TD decomposition with HAC on KuaiRand dataset. X-axis correspond to the learning rate for $Q$ learning.**

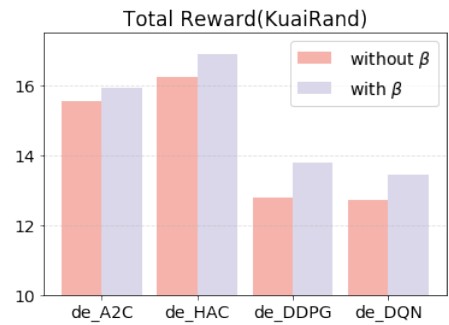

**Figure 8: Performance of stepwise TD approach with respect to $\beta$ in KuaiRand dataset.**

**Table 3: Comparison between past policy (upon sampling) and the current policy under different exploration magnitude. $\beta$ represents the debias term in $\mathcal{L}_{\beta-\text{stateTD}}$. $\alpha$ represents the absolute difference between action likelihood of the past and the present.**

|  | $\sigma$ | 1 | 0.9 | 0.7 | 0.5 | 0.3 | 0.1 | 0.01 |
|---|---|---|---|---|---|---|---|---|
| $\beta$ | ML1M | 0.96 | 0.96 | 0.94 | 0.89 | 0.76 | 0.37 | 0.04 |
|  | KuaiRand | 1.00 | 1.00 | 1.00 | 0.99 | 0.94 | 0.62 | 0.10 |
|  | Amazon | 0.99 | 0.98 | 0.98 | 0.97 | 0.96 | 0.80 | 0.10 |
| $\alpha$ | ML1M | 8e-4 | 1e-3 | 2e-3 | 9e-3 | 8e-2 | 3.55 | 7.4e2 |
|  | KuaiRand | 5e-6 | 8e-6 | 3e-5 | 4e-4 | 1e-2 | 1.47 | 6.5e2 |
|  | Amazon | 2e-4 | 4e-4 | 8e-4 | 3e-3 | 1e-2 | 0.59 | 6.5e2 |

*4.3.2 The Robustness under Action Exploration.* As we have discussed in section 3.3, the TD decomposition requires less sample to achieve accurate value function estimation and makes it easier for action exploration. To verify this claim, we compare the impact of different variances $\sigma$ in policy action exploration and summarize the results of the best model HAC (with decomposition) in Table 2. We can see that TD decomposition consistently outperforms the original TD learning across all settings of $\sigma$ and appears to be more robust to the action exploration. Specifically, in the KuaiRand environment, the original TD crashes when the exploration magnitude increases to $\sigma > 0.1$, but the TD decomposition still achieves accurate RL with even more improvement in the recommendation performance, corresponding to the Figure 1. In the ML1M environment, the TD decomposition achieves remarkable stability even when the exploration magnitude reaches $\sigma = 1$, while the original TD gradually deteriorates.

*4.3.3 Debased StateTD Learning and Stability.* To validate the effectiveness of the debias term in $\mathcal{L}_{\beta-\text{stateTD}}$ we conduct an ablation study that removes $\beta$ during learning on all four RL methods of TD decomposition. The results are summarized in Figure 8. We can see that the removal of the debias term of $\beta$ may generate suboptimal performance across all methods on both environments. Additionally, we investigate the $\beta$ term and the action discrepancy (mentioned in section 3.4) under different action exploration by changing the magnitude of $\sigma$. Specifically, we observe the TD decomposition in the best backbone method HAC and Table 3 shows this comparison in terms of $\beta$ and the average absolute difference $\alpha = |\pi(a_t|s_t) - p(a_t|s_t)|$. We can see that a larger exploration magnitude would end up with a closer distribution between the past and present, and the current policy has a higher chance of generating actions in the past (i.e. larger $\beta$ and smaller $\alpha$). Note that most baseline methods achieve the best results with small $\sigma$ in action exploration so that they can adapt to better states and actions. In contrast, TD decomposition achieves the same level of performance even with larger $\sigma$ which indicates that it works well for both stochastic policies and deterministic policies.

## 5 CONCLUSION

In this paper, we focus on the reinforcement learning methods that adopt temporal difference (TD) learning in recommender systems. We address the challenge of mixing random factors from stochastic policy and uncertain user environment and show that the traditional TD learning for long-term reward estimation is suboptimal or misguided. To achieve a more accurate approximation, we propose to engage a decomposed TD learning that eliminates the irrelevant random factors for each part and separates the approximation of $V$ and $Q$. The resulting framework achieves better recommendation performance, a faster learning process, and improved robustness against action exploration. While the proposed TD decomposition focuses on the value function learning which indirectly affects the policy learning in many RL methods, we believe that the investigation of the interactions between the value function and the actor may provide new perspectives on the interplay between users and the recommender system. In addition, our experiments and analysis originally emerged from the stochastic natural in recommendation problems, but we note that the proposed decomposition method can potentially be generalized to other RL problems as long as TD-based RL is adopted.

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

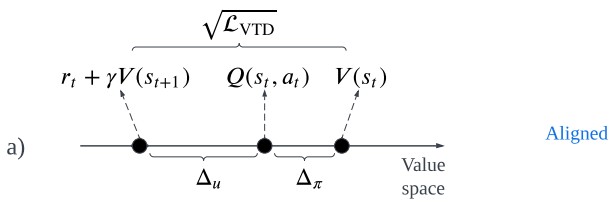

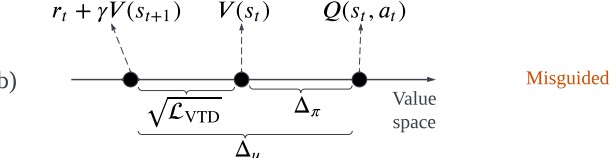

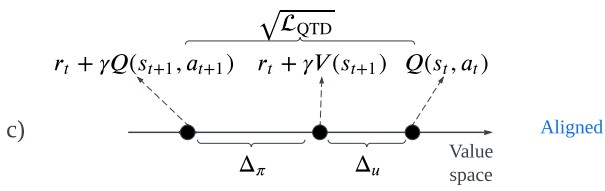

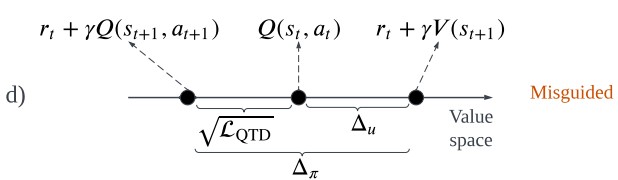

**Figure 9: Different cases of error and the corresponding relationship between $V$ and $Q$. The original TD loss may misguide the value estimation, while the decomposed method always minimizes the original TD.**

## A  RELATION WITH STEPWISE TD LEARNING

Suppose that the two objectives achieve a certain degree of accuracy with $\mathcal{L}_{\text{actionTD}} < \delta_1^2$ and $\mathcal{L}_{\text{stateTD}} < \delta_2^2$ for some small constants $\delta_1$ and $\delta_2$. Note that the action TD loss is semantically equivalent to the random user error $\Delta_u$ and the state TD loss is equivalent to the random policy error $\Delta_\pi$. Then, we can also state $\Delta_u < \delta_1$ and $\Delta_\pi < \delta_2$. And we can derive that the original stepwise TD loss is also bounded as $\mathcal{L}_{\text{VTD}} < (\delta_1 + \delta_2)^2$ in the worst case scenario.

To further investigate the relationship among $\mathcal{L}_{\text{actionTD}}$, $\mathcal{L}_{\text{stateTD}}$, $\mathcal{L}_{\text{VTD}}$, and $\mathcal{L}_{\text{QTD}}$, we analyze the common cases in Figure 9. Without loss of generality, in case a) where $Q$ is in between the two consecutive $V$s or case c) where $V$ is in between the two consecutive $Q$s, optimizing the stepwise TD loss $\mathcal{L}_{\text{VTD}}$ and $\mathcal{L}_{\text{QTD}}$ would also minimize $\Delta_\pi + \Delta_u$, which indirectly optimizes $\mathcal{L}_{\text{actionTD}} + \mathcal{L}_{\text{stateTD}}$. We consider these two cases as aligned cases where the original TD loss and the decomposed loss agree with each other. However, in case b) where consecutive $V$s locate on the same side of $Q$ or in case d) where consecutive $Q$s locate on the same side of $V$, minimizing the original stepwise TD loss no longer guarantees the correct minimization of $\mathcal{L}_{\text{actionTD}}$ and $\mathcal{L}_{\text{stateTD}}$. For example, case b) may trivially learn the bias of all states, so that the error between the two consecutive states' $V$ approaches zero, while the error of the policy's effect $\Delta_\pi$ and that of the user's randomness $\Delta_u$ are significantly larger. This further explains why stepwise TD is suboptimal under the mixing of random factors. In contrast, minimizing $\mathcal{L}_{\text{actionTD}}$ and $\mathcal{L}_{\text{stateTD}}$ guarantees a bounded minimization of the original TD loss for all four cases.

## B  FULL RESULTS OF MAIN EXPERIMENTS

Tables 4 and 5 present the depth performance results from the main experiments and the action exploration experiments.

## C  TRAINING CURVES FOR LONGER STEPS

We extend the number of online learning steps to 80,000 to further illustrate the converged performance, as shown in Figure 10.

## D  PERFORMANCE FOR DIFFERENT LEARNING RATE

We provide the extended comparison results of the learning rate effect in Figure 11.

**Table 4: The depth of action exploration in HAC.**

| $\sigma$ | Total Reward in ML1M | | Total Reward in KuaiRand | | Total Reward in Amazon | |
|---|---|---|---|---|---|---|
| | Original | Decomposed | Original | Decomposed | Original | Decomposed |
| 1 | 13.13 ±(4.51) | **18.04** ±(0.39) | 5.68 ±(0.10) | **10.25** ±(0.18) | 4.11 ±(0.16) | **4.77** ±(0.23) |
| 0.9 | 13.58 ±(4.54) | **18.01** ±(0.29) | 5.77 ±(0.19) | **10.64** ±(0.26) | 4.09 ±(0.11) | **4.87** ±(0.14) |
| 0.7 | 14.32 ±(4.43) | **17.78** ±(0.58) | 5.84 ±(0.23) | **12.96** ±(0.12) | 4.17 ±(0.26) | **5.44** ±(0.30) |
| 0.5 | 14.90 ±(3.73) | **17.94** ±(0.24) | 6.69 ±(0.72) | **15.62** ±(0.42) | 4.21 ±(0.19) | **6.57** ±(0.40) |
| 0.3 | 16.15 ±(2.16) | **17.95** ±(0.50) | 8.03 ±(0.63) | **16.75** ±(0.21) | 5.03 ±(0.00) | **10.89** ±(0.58) |
| 1e-1 | 17.35 ±(1.54) | **18.10** ±(0.36) | 11.55 ±(0.91) | **17.35** ±(0.45) | 8.72 ±(0.71) | **14.00** ±(0.73) |
| 1e-2 | 17.49 ±(0.88) | **17.76** ±(0.81) | 13.59 ±(0.84) | **16.72** ±(0.50) | 13.33 ±(0.53) | **14.31** ±(0.87) |

**Table 5: The depth of all methods and their correspongding decomposition.**

| Model | Depth in ML1M | | Depth in KuaiRand | | Depth in Amazon | |
|---|---|---|---|---|---|---|
| | Original | Decomposed | Original | Decomposed | Original | Decomposed |
| Non-RL | 16.55 ±(1.90) | - | 11.75 ±(3.17) | - | - | - |
| Dueling DQN | 16.45 ±(0.26) | - | 12.17 ±(3.69) | - | 11.88 ±(2.01) | - |
| A2C | 17.61 ±(0.29) | **17.97** ±(0.20) | 13.10 ±(0.76) | **16.51** ±(0.31) | 12.53 ±(0.66) | **14.13** ±(0.78) |
| DQN | 16.55 ±(0.45) | **16.71** ±(0.36) | 12.14 ±(3.94) | **14.41** ±(1.19) | 11.79 ±(2.22) | **13.12** ±(1.20) |
| DDPG | 14.47 ±(1.55) | **17.49** ±(0.87) | 13.92 ±(1.41) | **14.69** ±(1.35) | 12.08 ±(1.81) | **13.05** ±(1.98) |
| HAC | 17.35 ±(1.55) | **18.10** ±(0.36) | 13.59 ±(0.84) | **17.35** ±(0.45) | 13.33 ±(0.53) | **14.31** ±(0.87) |
| SQN | 16.33 ±(0.45) | **16.88** ±(0.38) | 11.22 ±(0.76) | **15.42** ±(0.70) | 8.05 ±(0.45) | **12.95** ±(0.71) |

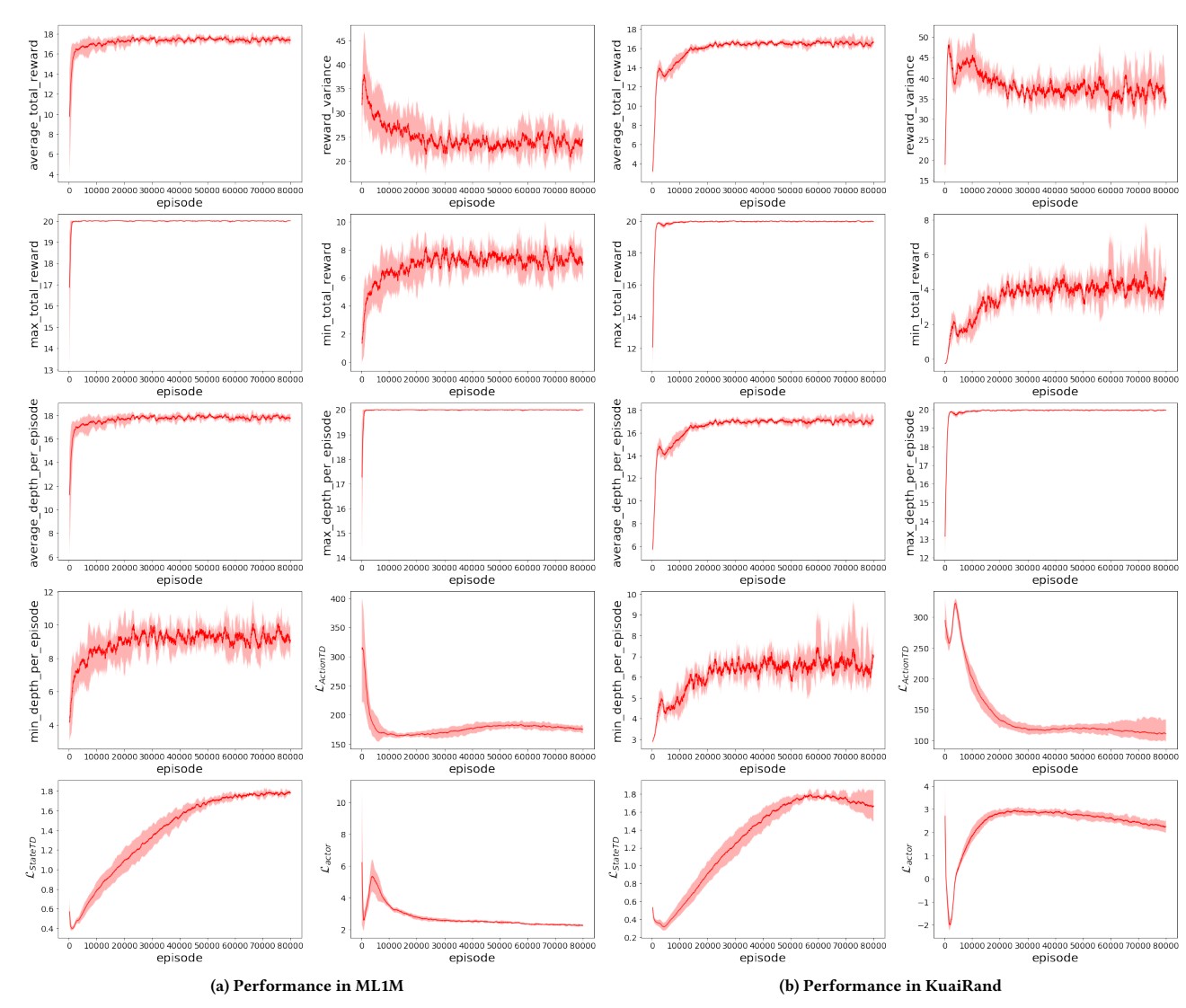

(a) Performance in ML1M

(b) Performance in KuaiRand

**Figure 10: Longer steps for Training curves of TD Decomposition with HAC.**

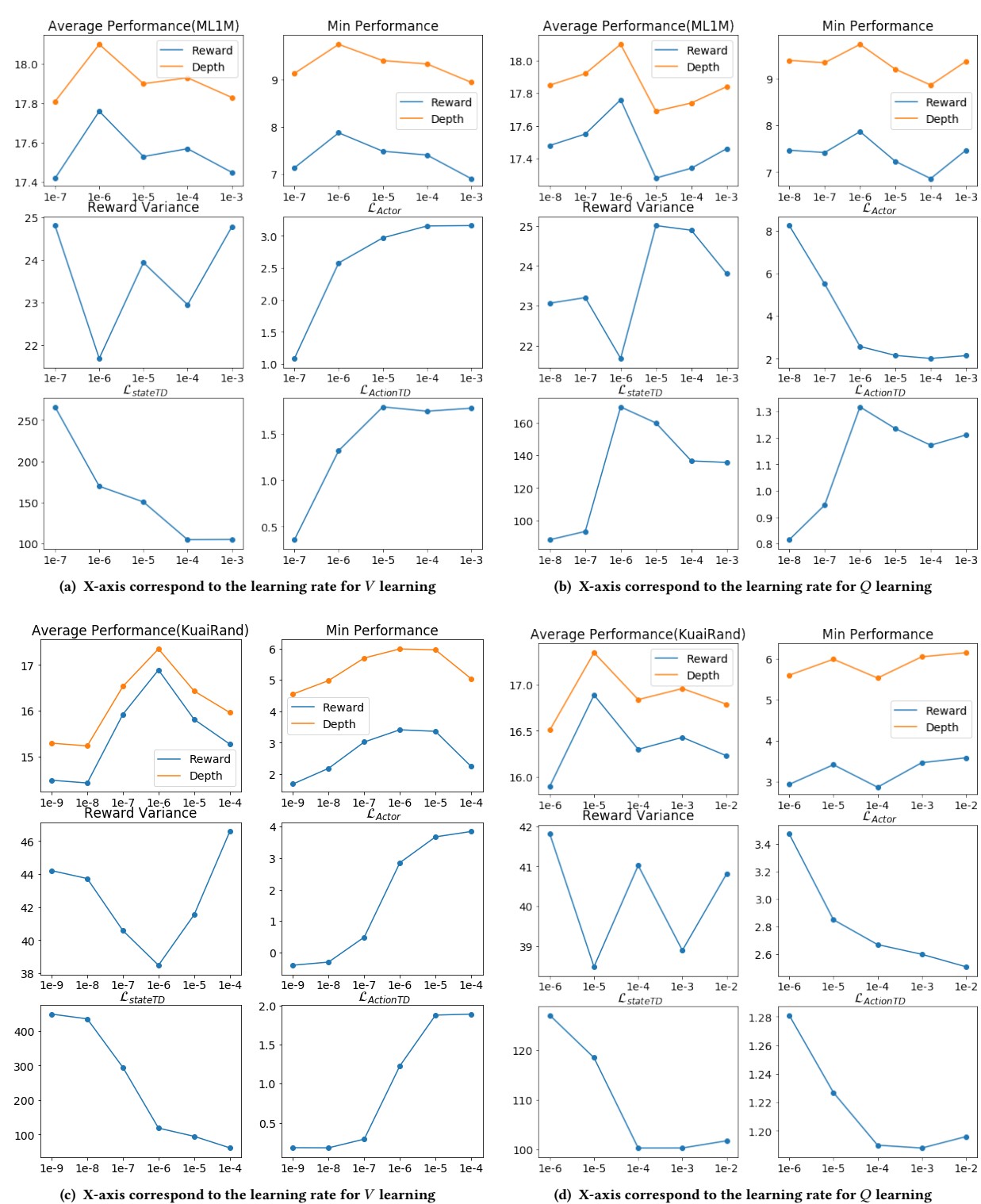

(a) X-axis correspond to the learning rate for $V$ learning

(b) X-axis correspond to the learning rate for $Q$ learning

(c) X-axis correspond to the learning rate for $V$ learning

(d) X-axis correspond to the learning rate for $Q$ learning

Figure 11: The effect of TD decomposition with HAC , where (a) and (b) denote the ML1M dataset, and the others correspond to KuaiRand dataset .

