# OpenReview forum: "Value Function Decomposition in Markov Recommendation Process"
_ACM.org/TheWebConf/2025/Conference — WWW 2025 Oral_

### Official Review · Reviewer_Bcp9 · 2024-12-02

**Novelty:** 4
**Technical Quality:** 4

**Review:**

This manuscript proposes a decomposed temperal difference (TD) learning framework to address the challenge of suboptimal TD learning.

*Pros*
- Reinforcement learning for recommendation is an interesting research topic.

*Cons*
- The related work section needs more discussions on the existing studies that address the suboptimal TD challenge.
- Although the authors claim that implementation details can be found in their released codes, important implementation details shall be provided in the manuscript to improve readability.
- More details on the simulated environment for online learning shall be provided, e.g., how the simulated environment resemble the real one?
- Some writing issues need to be fixed. For example, in Abstract, the acronym “TD” is used without a prior explanation. In the title of Table 1, the word "correspongding" shall be a typo.

**Questions:**

- The related work section needs more discussions on the existing studies that address the suboptimal TD challenge.
- Although the authors claim that implementation details can be found in their released codes, important implementation details shall be provided in the manuscript to improve readability.
- More details on the simulated environment for online learning shall be provided, e.g., how the simulated environment resemble the real one?
- Some writing issues need to be fixed. For example, in Abstract, the acronym “TD” is used without a prior explanation. In the title of Table 1, the word "correspongding" shall be a typo.

**Reviewer Confidence:**

2: The reviewer is willing to defend the evaluation, but it is likely that the reviewer did not understand parts of the paper

**Scope:**

4: The work is relevant to the Web and to the track, and is of broad interest to the community

---

### Official Review · Reviewer_SQ1E · 2024-12-02

**Novelty:** 5
**Technical Quality:** 6

**Review:**

Overview:
This paper studies the online reinforcement learning for recommender system, which is an important topic that formulates the user-system interaction as long-term optimization problem. In this work, the authors reveal the incorrect approximation issue in existing TD learning. To address the issue, this paper introduces a decomposition approach, which separately approximates the user state’s long-term utility and stochastic user environment. In addition, the authors introduce an extra debias term to help refine the learning of V for a correct estimation. Extensive experiments on three datasets validate the effectiveness of the proposed method.

Pros:
- This work is well-motivated. The identified incorrect approximation from the mixing random factors in TD learning is interesting and rational.
- The proposed method — both the decomposed modeling and the introduced debias term is intuitive and novel.
- Substantial experiments on three datasets under various settings validate the effectiveness of the proposed method.
- The paper is well-organized and easy to follow.


Several minor suggestions:
- Since the scope is restricted to TD learning, it would be better if a more comprehensive comparison and discussion with other RL approaches could be included, at least in related work. Currently, the reason for choosing TD is simply “due to its stepwise learning framework that well-suits the recommendation task …”. A detailed explanation or at least some supporting reference will be more convincing and solid.
- It would be great if more intuitive explanations or examples could be included, which could help user understand the challenge of mixed randomness in existing methods.

**Questions:**

Please refer to minor suggestions.

**Reviewer Confidence:**

2: The reviewer is willing to defend the evaluation, but it is likely that the reviewer did not understand parts of the paper

**Scope:**

4: The work is relevant to the Web and to the track, and is of broad interest to the community

---

### Official Review · Reviewer_Qk45 · 2024-12-04

**Novelty:** 5
**Technical Quality:** 5

**Review:**

This paper studies the decomposition of the TD learning process for a robust value function learning of RL policies. Specifically, by regressing the V-function on the Q-function and Q-function on the V-function (instead of using only one of Q-function or V-function), the TD learning can separate the noise caused by stochastic action choice and state transition, resulting in improved performance of the TD learning and corresponding policy.

---

**Technical quality**

- **Simple yet effective, sound idea**: Decomposing the TD learning process into two phases is quite reasonable given the connection between Q-function and V-function.
- **Good experiment results**: The benefit of using the decomposed TD-learning is shown on several semi-synthetic environments based on open-source recommendation datasets with many different base RL algorithms, and the results seem promising. The paper also tests the proposed methods with many different learning rates.

---

**Clarity**

- The paper is well-written, easy-to-follow.
- Minor typos: page 1, line 100, user`s; page 4, line 405, $\delta$ (instead of $\Delta$)

---

**Originality and significance**

- I believe the decomposition of TD-learning is new to the RL community.

---

**Weaknesses**

- **Assumption of observing the state transition**: One concern I have is whether assuming the observation of user states is reasonable in recommender systems. I may have missed the definition of the user states in the paper, but usually, accessing the user state that can fully explain the changes in user response is challenging. Many learning-to-rank papers simply add the observed items as a part of the state (e.g., Zhang et al., 2023).

---

**References**

- Zhang et al., 2023: Zeyu Zhang, Yi Su, Hui Yuan, Yiran Wu, Rishab Balasubramanian, Qingyun Wu, Huazheng Wang, Mengdi Wang. Unified Off-Policy Learning to Rank: a Reinforcement Learning Perspective. NeurIPS2023.

**Questions:**

- How does the experiment model the definition and transition of the states?

- How does the baseline work with $\beta$? I was wondering if the gain comes from importance sampling or the decomposition of the TD-learning loss.

- How do the conventional VTD and QTD differ between the Q and V functions learned by the naive training and the proposed decomposition. The figure of action-TD and state-TD is informative, but I think the comparison of VTD and QTD is more evident to show the benefit of the proposed method.

- What is the implication of Figure 5? The compared algorithms behave differently between action-TD and state-TD, and what are the insights from these observations?

**Reviewer Confidence:**

3: The reviewer is confident but not certain that the evaluation is correct

**Scope:**

4: The work is relevant to the Web and to the track, and is of broad interest to the community

---

### Official Review · Reviewer_umbn · 2024-12-04

**Novelty:** 6
**Technical Quality:** 5

**Review:**

The authors proposed in this paper to decompose the value function, which is used to estimate user rewards in online reinforcement learning, into two separate factors and to approximate the two factors separately during the learning process via a disentangled learning framework to achieve more accurate and faster learning with improved robustness against action exploration. This work belongs to Recommender Systems studies and fits the scope of World Wide Web research.

Obtaining a stable and accurate value function has been a longstanding issue in online reinforcement learning. This work innovatively considers the factors from two perspectives: random action exploration and stochastic user environment. The idea appears novel to me.

Overall, the paper is well presented, with formal problem definitions and method descriptions. The proposed approach looks sound and easy to follow.

**Questions:**

none

**Reviewer Confidence:**

3: The reviewer is confident but not certain that the evaluation is correct

**Scope:**

4: The work is relevant to the Web and to the track, and is of broad interest to the community

---

### Official Review · Reviewer_hJXf · 2024-12-05

**Novelty:** 4
**Technical Quality:** 3

**Review:**

This paper proposes a Markov recommendation method that separately approximates two factors from the stochastic policy and the uncertain user environment by decomposing the original temporal difference loss. I think there are some aspects that need to be improved.

1. This paper argues that key reason that limit the accuracy of the value function is the mixed view of the two random factors. However, there is no detailed explanations for this statement in either Abstract or Introduction. It is recommended to add more detailed descriptions in proper positions.

2. It is suggested to reorganize the related work, which only contains one aspect in current version. It is hard to learn current research progress in the related field from only one aspect of related work.

3. The baseline methods are outdated, with only one from recent three years. This cannot fully verify the effectiveness of the proposed method. It is preferable to add more latest and state-of-the-art competitive methods. Even worse, there is a baseline method that can be dated back to 1999. It is highly recommended to reconsider whether such an old baseline is necessary to be included.

4. The hyperparameter analysis in the paper predominantly emphasizes the investigation of the learning rate, which is insufficient. Several critical hyperparameters, such as the discount factor in Eq. (1), have not been studied in experimental analysis, limiting the comprehensiveness of the study.

5. It is preferable to explain why the decomposed performance of Non-RL and Dueling DQN cannot be obtained.

6. There are some grammar, typo and formatting problems, in line 99, 113, 755. Besides, all equations are ended with no punctuation, which is not up to the code. Additionally, the formatting of references is inconsistent in terms of abbreviation, capitalization issue, name of publishers, etc. It is recommended to check the whole paper thoroughly to correct these problems.

**Questions:**

Why is the original performance of Non-RL on the Amazon dataset omitted in Table 1 and Table 5?

**Reviewer Confidence:**

3: The reviewer is confident but not certain that the evaluation is correct

**Scope:**

3: The work is somewhat relevant to the Web and to the track, and is of narrow interest to a sub-community

---

### Official Review · Reviewer_c3hg · 2024-12-09

**Novelty:** 5
**Technical Quality:** 4

**Review:**

The paper maily focuses on RL for recommendation. The contribution is the decomposition of value functions. The author attributes the variance of value function into policy randomness and user environment dynamics. The author proposed two loss functions to downgrade the randomness. Experimental results demonstrate the effectivess of the proposed methods.
Pros:
1. The paper is well-written and easy to follow.
2. The idea to decompose value function into user dynamic and policy randomness is interesting.
3. Experiments demonstrate the effectivess of the method.
Cons:
Please see the questions

**Questions:**

1. I'd like to know how the user similator is trained on the offline dataset. How to verify that the trained simulator can effectively mimic user environment.
2. Some details should be further examined to make sure the correctness. For example, Eq.(5) indeed describes the loss function of SARA, not Q-learning.
3. What is the K used in this paper? Since the action space is I^{K}, leading to the exponentially increased space. How the author address the large action space. Besides, datasets like ML and Amazon actully contains point-wise ground-truth, not list-wise  I^{K}. So how the author address such above problem.

**Reviewer Confidence:**

4: The reviewer is certain that the evaluation is correct and very familiar with the relevant literature

**Scope:**

4: The work is relevant to the Web and to the track, and is of broad interest to the community